# *Clostridioides difficile* Infection Treatment and Outcome Disparities in a National Sample of United States Hospitals

**DOI:** 10.3390/antibiotics11091203

**Published:** 2022-09-06

**Authors:** Eric H. Young, Kelsey A. Strey, Grace C. Lee, Travis J. Carlson, Jim M. Koeller, Kelly R. Reveles

**Affiliations:** 1College of Pharmacy, The University of Texas at Austin, Austin, TX 78701, USA; 2Pharmacotherapy Education and Research Center, University of Texas Health San Antonio, San Antonio, TX 78229, USA; 3Fred Wilson School of Pharmacy, High Point University, High Point, NC 27268, USA

**Keywords:** *Clostridioides difficile*, epidemiology, disparities, mortality

## Abstract

*Clostridioides difficile* infection (CDI) disproportionately affects certain populations, but few studies have investigated health outcome disparities among patients with CDI. This study aimed to characterize CDI treatment and health outcomes among patients by age group, sex, race, and ethnicity. This was a nationally representative, retrospective cohort study of patients with laboratory-confirmed CDI within the Premier Healthcare Database from January 2018 to March 2021. CDI therapies received and health outcomes were compared between patients by age group, sex, race, and Hispanic ethnicity using bivariable and multivariable statistical analyses. A total of 45,331 CDI encounters were included for analysis: 38,764 index encounters and 6567 recurrent encounters. CDI treatment patterns, especially oral vancomycin use, varied predominantly by age group. Older adult (65+ years), male, Black, and Hispanic patients incurred the highest treatment-related costs and were at greatest risk for severe CDI. Male sex was an independent predictor of in-hospital mortality (aOR 1.17, 95% CI 1.05–1.31). Male sex (aOR 1.25, 95% CI 1.18–1.32) and Black race (aOR 1.29, 95% CI 1.19–1.41) were independent predictors of hospital length of stay >7 days in index encounters. In this nationally representative study, CDI treatment and outcome disparities were noted by age group, sex, and race.

## 1. Introduction

The inequitable distribution of disease in the United States (U.S.) is a reflection of the intricate relationship between patient characteristics, such as racial or ethnic group, religion, socioeconomic status, gender, and others impacted by discrimination, and the resulting obstacles that prevent people from achieving high quality care and good health [1]. Though efforts are growing to characterize and eliminate health disparities, as demonstrated by the Healthy People 2020 goals for health disparities, the U.S. population continues to rapidly change resulting in more individuals comprising groups with historically worse health access and outcomes. In 2050, older adults (65 years and older), Hispanic people and Black people are projected to comprise a larger proportion of the population compared to 2005 [2]. Health disparities may ultimately lead to inequity in incidence, prevalence, mortality, and financial healthcare burden [1]. It was estimated that eliminating minority disparities in healthcare could reduce direct medical care expenditures by USD 230 to USD 243 billion per year [3,4]. Infection is the second leading contributor to racial disparities in all-cause mortality [5]. A growing body of literature is dedicated to investigating the complicated relationship between common infectious diseases, such as the human immunodeficiency virus (HIV) and the coronavirus disease 2019 (COVID-19), and certain patient populations; however, studies examining *Clostridioides difficile* infections (CDI) are limited [6,7,8,9].

CDI affects nearly half a million Americans annually and results in substantial healthcare burden, including poor patient quality of life, lengthy hospital stays, risk for recurrent infection, and high treatment costs [10,11]. The incidence and health outcomes associated with CDI may differ based on age group, sex, race, and ethnicity due to a myriad of factors, including multimorbidity, health care exposures, insurance coverage, access to care, quality of care, socioeconomic factors, environment, and microbiome composition [12,13,14]. CDI is a major problem, specifically in older adults, with one out of every three CDIs occurring in patients aged 65 years or older [15,16]. In addition to being at higher risk for infection, older adults experience the longest median hospital stays compared to younger adults and children [17]. Additionally, studies have shown that older adults as well as females are at a higher risk for CDI and recurrence [18,19]. Disparities in CDI also appear to exist among racial and ethnic groups. Multiple studies have found that CDI incidence was higher in White patients compared to non-White or Black patients [7,9,19]. Despite a lower incidence, Black patients have been found to have poorer health outcomes compared to White patients, including longer hospital length of stays (LOS), higher recurrence rates, and greater risk for severe CDI and mortality [9,20].

While studies have focused on CDI-related outcomes in the overall and specific patient populations, few studies have evaluated how patient characteristics may influence these outcomes on a national level. Therefore, the primary objective of this study was to identify CDI treatment and health outcomes disparities by age group, sex, race, and ethnicity.

## 2. Results

### 2.1. Population Characteristics

Using administrative codes only, a total of 278,019 unique CDI encounters were identified. Of these, 45,331 encounters included laboratory-confirmed CDI, representing 38,764 index encounters and 6567 encounters for presumed recurrent CDI (rCDI). CDI patient characteristics for the index and recurrent encounters are provided in Table 1. Patients in both index and recurrent encounters were generally older (median age 68 and 69 years, *p* = 0.009), White (81.1% vs. 81.9%, *p* = 0.003), non-Hispanic (94.4% vs. 95.2%, *p* = 0.023), and female (57.6% vs. 62.1%, *p* < 0.001). Overall, the most common payment source for index and recurrent encounters was Medicare (64.5% and 69.5%). Additionally, most patients required an inpatient admission (87.2% vs. 80.9%, *p* < 0.001) that was considered emergent (78.3% and 77.4%).

### 2.2. Disparities in CDI Treatment and Treatment-Related Costs

Notable differences were seen in CDI treatments administered and costs incurred by patient subgroup for index encounters (Table 2). Oral vancomycin was the most common CDI treatment administered overall, but it was significantly more common among patients aged 18 to 64 years and 65 years and older compared to those 18 years and younger (74.8% and 76.2% vs. 37.9%, *p* < 0.001), males compared to females (76.7% vs. 74.0%, *p* < 0.001), patients of other race compared to Black and White individuals (77.6% vs. 74.0% vs. 75.3%, *p* = 0.003), and Hispanic compared to non-Hispanic patients (78.0% vs. 74.9%, *p* = 0.003). Overall, fidaxomicin use was low, particularly among patients aged 18 years and younger (0.1%). Fecal microbiota transplantation (FMT) was also uncommon among all patient subgroups, but its use was highest among Hispanic patients compared to non-Hispanic patients (0.4% vs. 0.1%, *p* = 0.010). When comparing age groups, patients aged 65 years and older incurred the highest hospital-associated costs and patient-associated charges (USD 11,811 and USD 45,333) compared to patients 18 to 64 years of age (USD 10,508 and USD 42,612) and patients less than 18 years of age (USD 5407 and USD 16,249) (*p* < 0.001 for both). Costs and charges were highest among patients belonging to other races (USD 15,162 and USD 56,681, respectively) and Black patients (USD 13,985 and USD 53,811, respectively) compared to White patients (USD 10,672 and USD 41,817, respectively) (*p* < 0.001 for both). Male patients also incurred higher costs and charges (USD 13,586 and USD 52,321) compared to females (USD 9999 and USD 39,387) (*p* < 0.001 for both). Lastly, Hispanic patients incurred slightly higher costs and charges (USD 11,154 and USD 47,957) compared to non-Hispanic patients (USD 10,959 and USD 42,888) (*p* = 0.013 and *p* < 0.001).

Similar disparities were seen in CDI treatments administered and costs incurred by patient subgroup for recurrent encounters (Table 3). Oral vancomycin use for recurrence was higher in Hispanic compared to non-Hispanic, male compared to female, and other race and Black compared to White patients. Overall, fidaxomicin use was still low, particularly among patients aged 18 years and younger (1.6%), though the frequency of use was slightly higher in all patient subgroups in comparison to the use of fidaxomicin in index encounters. While FMT was still uncommon among all patient subgroups, it was utilized more in recurrent cases. Costs and charges were similarly the highest among male compared to female, other race and Black individuals compared to White individuals, and Hispanic patients compared to non-Hispanic patients. Interestingly, hospital-associated costs and patient-associated charges were overall numerically lower across all patient subgroups compared to index visits.

### 2.3. Disparities in CDI Patient Health Outcomes

Comparisons of CDI patient health outcomes by subgroup for index encounters are provided in Table 4. Disparities were observed between groups in the risk for severe CDI; increased risk was found among adults 65 years and older and adults aged 18 to 64 years compared to those less than 18 years of age (aOR 1.99, 95% CI 1.40–2.81 and aOR 1.80, 95% CI 1.28–2.54), males compared to females (aOR 1.45, 95% CI 1.37–1.53), Black compared to White patients (aOR 1.85, 95% CI 1.71–2.00), patients of other race compared White patients (aOR 1.26, 95% CI 1.12–1.41), and Hispanic compared to non-Hispanic patients (aOR 1.22, 95% CI 1.08–1.38). Patients aged 65 years and older (aOR 2.07, 95% CI 1.11–3.88), aged 18–64 years (aOR 2.01, 95% CI 1.09–3.73), and Black patients (aOR 1.17, 95% CI 1.03–1.34) were independent predictors of CDI recurrence. All-cause, in-hospital mortality and hospital LOS were numerically higher and statistically different in bivariate analyses for patients 65 years and older and 18 to 64 years of age, males, and Black and other races compared to their reference groups (all *p* < 0.05). However, only male sex was independently associated with mortality (aOR 1.17, 95% CI 1.05–1.31) after adjusting for age, race, and other covariates, including severity. Being Black (aOR 1.29, 95% CI 1.19–1.41) and of the male sex (aOR 1.25, 95% CI 1.18–1.32) were independently associated with prolonged hospital LOS once covariates were controlled for. Hispanic ethnicity was not associated with significantly higher frequency of mortality or hospital LOS (both *p* > 0.05).

Comparisons of CDI patient health outcomes by subgroup for recurrent encounters are shown in Table 5. Disparities were observed between groups in the risk for severe CDI, in which increased risk was found among adults 65 years and older and adults aged 18 to 64 years compared to those less than 18 years of age (aOR 11.98, 95% CI 1.49–96.26 and aOR 9.65, 95% CI 1.21–77.06), males compared to females (aOR 1.62, 95% CI 1.42–1.86), Black compared to White patients (aOR 2.12, 95% CI 1.75–2.56), and Hispanic compared to non-Hispanic patients (aOR 1.98, 95% CI 1.43–2.73). However, it is important to note the relatively wide 95% confidence intervals corresponding to severe CDI risk, which makes these estimates imprecise. Similar to index encounters, all-cause mortality was numerically higher and statistically different in bivariate analysis for patients 65 years and older and 18 to 64 years of age; however, no variables were found to be independently associated with mortality after adjusting for covariates. Male sex and Black race were independent predictors of prolonged hospital LOS. Finally, a summary of patient characteristics and health outcomes by study year can be found in the Appendix A. Overall, patient characteristics were similar between each year. Vancomycin use increased and metronidazole use decreased each year. Costs and charges were highest in 2020 and 2021. Lastly, CDI outcomes were similar across years as well, though median hospital LOS was longer and mortality was higher in 2021.

## 3. Discussion

In this nationally representative study of U.S. hospitals, there were notable differences in treatment characteristics between CDI patients of different age groups, sexes, races, and ethnicities. Additionally, certain patient populations were at significantly higher risk for poor health outcomes, including severe CDI, mortality, and prolonged hospital stays.

This is one of the first studies to evaluate CDI treatment disparities by patient subgroup. While a direct comparison between the full U.S. population and the patient population in this study is not possible, patient characteristics for both index and recurrent encounters differed from the general U.S. population. For example, compared to the general U.S. population, CDI patients in this study were more often age 65 years or older (15% vs. 58%), White (75% vs. 81%), non-Hispanic (82% vs. 94%), and female (50% vs. 58%) [21]. Furthermore, patient characteristics for both index and recurrent encounters were similar to characteristics presented in other nationally representative CDI studies utilizing the CDC’s U.S. National Hospital Discharge Surveys [9,17]. We noted that oral vancomycin use was the most common CDI treatment, particularly among adults (age 18–64) and older adults (65 years and older), while metronidazole was most common among pediatric patients in index encounters. These trends are in-line with prior CDI clinical practice guidelines that include pediatric recommendations. In the newest CDI guideline update, fidaxomicin was recommended as first-line therapy for adult populations and bezlotoxumab was recommended for certain high-risk populations [22]. Notably, the use of both of these agents was low in this study, particularly during index visits, which is unsurprising given the relative higher costs and rather recent guideline update; however, fidaxomicin and FMT use was higher during recurrent encounters. These findings reflect additional need for resources to support the widespread clinical implementation of these first-line therapies.

While we did not find large absolute differences in the percentage of patient populations receiving certain CDI therapies, we did find large disparities in hospital-associated costs and patient-associated charges in both index and recurrent encounters. For example, our study showed that the median hospital-associated cost was numerically highest for patients of other races, followed by Black patients, compared to White patients, male compared to female patients, and patients 65 years and older compared to patients patient less than 18 of age. Median patient-associated charges followed a similar pattern. These costs and charges could be attributed to several factors, including longer hospital stays and more severe CDI in these populations. Antibiotic and other supportive treatment (e.g., surgery) and other underlying comorbidities during their initial episode could have also affected cost. While few studies have investigated cost disparities, our findings are consistent with attributable CDI costs reported in several studies measuring the overall economic burden of CDI. In a meta-analysis by Ghantoji et al., the incremental cost of CDI ranged from USD 2871 to USD 90,664, the higher range included patients in special populations, (e.g., patients with irritable bowel disease, surgical patients, critically ill patients) [23]. A subsequent systematic review described the mean attributable CDI costs to range from USD 8911 to USD 30,049 for hospitalized patients [24].

Unfortunately, we observed poorer health outcomes in certain racial and ethnic subgroups, which is similar to previous observations. In a national study of CDI patients by Argamany et al., mortality (7.4% vs. 7.2%) and median hospital LOS (9 vs. 8 days) were slightly higher in Black patients versus White patients, respectively, even though CDI incidence was higher in White patients (7.7 per 1000 total discharges) compared to Black patients (4.9 per 1000 total discharges) (*p* < 0.0001) [9]. Additionally, Black race was an independent predictor of severe CDI (OR 1.09, 95% CI 1.07–1.11, *p* < 0.0001) and mortality (OR 1.12, 95% CI 1.09–1.15, *p* < 0.0001). While the present study did not find that Black patients were at a higher risk of mortality, it did find that Black patients were at a significantly greater risk for severe CDI and longer hospital LOS compared to White patients. Furthermore, this study found that Hispanic patients were also at a significantly increased risk of severe CDI. However, there have been no previous studies examining differences in CDI patient health outcomes, treatment patterns, or costs between Hispanic and non-Hispanic patients.

Several factors, such as health insurance and quality of care, are likely contributors to these disparities. We observed that more Black patients relied on Medicaid compared to White patients, which is consistent with previous studies. For example, in 2016, the number of Black patients requiring government health insurance (i.e., Medicare and Medicaid) was 43.7% compared to Hispanic patients (40.1%) and non-Hispanic White patients (35.9%). Additionally, the number of uninsured was highest among Hispanic patients (16.0%) followed by Black patients (10.5%) compared to non-Hispanic White patients (6.3%) [25]. Interestingly, our study found Black race to be associated with patient health outcomes independent of payor status. This highlights the complex relationship between patient characteristics and other factors contributing to health disparities. In addition to health insurance coverage, studies have also shown that quality of care can significantly impact health outcomes. Although few studies have analyzed the direct association between these socioeconomic factors and CDI severity, there is evidence that racial and ethnic disparities exist in quality of care. A study by Fiscella et al. examined quality of care measures in the United States by utilizing the 2014 Quality and Disparities Reports compiled by the Agency for Healthcare Research and Quality (AHRQ) and the 2015 National Impact Assessment of the Centers for Medicare and Medicaid Services (CMS) Quality Measures Report. After identifying limitations in these reports, the authors identified common quality measures that included experience of care, preventive care, chronic disease control, hospitalizations, obstetrics, and behavioral health and examined published studies over a ten-year period. The combined findings in this study suggest that a large proportion of Black and Hispanic patients experience worse quality of care compared to White patients [26]. Racial and ethnic disparities in comorbid conditions may also contribute to worse CDI health outcomes. Multiple systematic reviews have confirmed the relationship between multimorbidity and increased mortality, decreased functional status, and decreased quality of life [27,28]. A study by Quiñones et al. analyzed middle-aged and older adults in the United States who responded to the Health and Retirement Study from 1998 to 2014 and found that Black patients initially have higher chronic disease counts than White patients (IRR 1.279, 95% CI 1.201–1.361). Furthermore, Hispanic patients were found to accumulate chronic disease 1.5% faster than White patients (IRR 1.015, 95% CI 1.001–1.028), though initially they had lower levels of chronic disease burden compared to White patients [29]. Disadvantages seen in minority racial and ethnic groups, like poverty, lower health literacy, distrust in the medical system that may lead to avoidance or delay in care, and higher rates of unemployment are also factors that, although were not analyzed in our study, are potential contributors that should be considered in future studies [30].

In addition to outcome disparities by race and ethnicity, differences were also seen in health outcomes when patients were stratified by sex. In our analysis, male patients had a longer median hospital LOS and higher mortality rates compared to female patients. Males were also more likely to have severe CDI, in-hospital mortality, and longer hospital LOS. Though all comparisons for in-hospital mortality were insignificant for recurrent encounters likely due to small sample size. Additionally, male sex was an independent predictor of mortality. While few studies have evaluated the direct effects of sex on CDI outcomes, there are several factors that could lead to increased morbidity and mortality in the male population. For example, in a Danish study by Höhn et al., researchers noted that between 2005 and 2014, women were more likely to experience a longer time to first hospital admission compared to men (10.3 vs. 9.4 years, respectively) [31]. Additionally, men are less likely to utilize primary healthcare services than women, which may lead to a delayed seeking of diagnosis and treatment and therefore potentially more severe disease states upon presentation at the hospital [32]. While there is scarce evidence on CDI mortality by sex, previous studies have shown that overall, hospitalized males have a higher mortality rate compared to females. For example, in another Danish study by Höhn et al., the one-year risk of mortality for all-cause hospitalized admissions was significantly higher for men at age 50 years (5.17%, 95% CI 4.60–5.73) compared to women of the same age (2.97%, 95% CI 2.66–3.29) [33]. As males have been shown to have more severe conditions, such as heart disease, stroke, and diabetes, these comorbidities can also play a significant role in infection severity and outcomes in patients who develop CDI [34].

Lastly, this study also noted significant disparities in CDI outcomes by age groups. For example, patients 65 years and older were at the highest risk for severe CDI. Older adults also had higher mortality and longer median hospital LOS compared to the younger patient groups, though age group did not remain a significant predictor of these outcomes once covariates, including severity, were controlled for. Furthermore, the disparity for severe CDI was more pronounced in recurrent encounters and hospital LOS was shorter in older adults compared to younger age groups, though the risk estimate was imprecise due to smaller sample sizes. The higher risk of severe CDI for older adults is consistent with previous studies, including a study evaluating CDI-associated deaths between 1999 and 2004, which showed that the median age of CDI mortality was 82 years [35]. As aging is a primary risk factor for the development of CDI, it is well known that comorbidities, prior hospitalizations and medications, and residence in a long-term care facility can further increase the risk of negative CDI outcomes in this vulnerable patient population. In addition to health care exposures, age-related physiologic changes also play a role in disparities. The gut microbiota, responsible for resisting the colonization of *C. difficile,* changes during the aging process. A study by Hopkins and Macfarlane found differences in bacterial composition between young healthy, older healthy, and patients with CDI [36]. As the human body ages, it is also accompanied by a gradual deterioration of the immune system known as immunosenescence. Decreased antibody production and protection against *C. difficile* toxins may not only increase disease severity but also recurrence and mortality [37,38].

There were several limitations to this study. First, due to its retrospective nature, diagnosis and treatment data may be subject to misclassification bias and confounding. To limit confounding, we used multivariable modeling with covariates that were likely to affect study outcomes; however, it cannot be determined from this model that outcomes were specifically attributable to CDI compared to other unmeasured variables. Second, race and ethnicity may have been self-reported or missing and individuals that did not fall into the Black or White race categories were classified as “other” due to smaller sample sizes. These could have limited the accuracy of our demographic classifications. Third, treatment choice and subsequent outcomes over the three-year time period may have been influenced by diagnostic stewardship over time and clinical guideline recommendations regarding oral vancomycin and metronidazole, with the latter falling out of favor in recent years. Fourth, we used a strict definition of recurrent CDI (i.e., laboratory-confirmed) and it is possible that some patients were lost to follow-up, both of which may have underestimated true recurrence rates. Fifth, large sample sizes increase the likelihood of demonstrating statistically significant differences between groups; absolute between-group differences must be considered in addition to relative differences and *p*-values. Sixth, misclassification bias may have occurred for children who may be colonized with *C. difficile* before the age of 5 years who may not have a true CDI. There were very few patients in this population subgroup younger than 5 years old, so all pediatric patients were grouped together. Finally, there are many factors that may have influenced CDI treatment and outcome disparities that we were unable to control for in this study. For example, this study was not able to account for potential microbiota differences between subpopulations and several socioeconomic variables, such as education, income status, and employment. The dataset did not include microbiota data and was deidentified, so it was not possible to acquire that information. These are all factors that could potentially further influence disparities in CDI-related outcomes.

## 4. Materials and Methods

### 4.1. Study Design and Data Source

This was a retrospective cohort study using data from the U.S. Premier Healthcare Database (PHD) from January 2018 to March 2021. The PHD contains robust and detailed data from approximately 1041 hospitals in the U.S., primarily from non-profit, non-governmental, community, teaching hospitals, and health systems [39]. This database includes over 230 million unique patient visits and contains information on hospital visits, both inpatient and outpatient, and events that occur during a patient’s visit, including drug administration, cost and charge data, and diagnoses. Additionally, the database uses a masked identifier that allows for tracking patients longitudinally across encounters. The PHD is considered exempt from institutional review board oversight.

### 4.2. Participants

CDI patients included in Premier were first identified if their visit (inpatient or outpatient) contained an International Classification of Diseases, 10th Revision (ICD-10 code) for CDI (A04.72). Patients with an ICD-10 code indicative of recurrent CDI (A04.71) at cohort entry were excluded. This population was then limited to those patients with laboratory-confirmed CDI (i.e., those with any positive *C. difficile* stool test (e.g., toxin enzyme immunoassay, glutamate dehydrogenase antigen, nucleic acid amplification test). The index encounter included only the first unique encounter for a patient confirmed with both an ICD-10 code and positive stool test. An encounter following the index was only included if it met the definition for recurrent CDI, otherwise duplicate encounters were excluded.

### 4.3. Outcomes

The primary outcome of this study was disparities in CDI treatment utilization and healthcare costs. Secondary outcomes included severity of CDI, recurrent CDI, hospital length of stay, and all-cause hospital mortality.

### 4.4. Variable Definitions

Patient baseline characteristics included age, sex, race, ethnicity, and payor type. Race (Black, other, White) and ethnicity (Hispanic, non-Hispanic) categories were based on the PHD terminology. The race category “other” was defined by the investigators and included patients categorized as Alaska Native, Asian, or other. The ethnicity category identifies those who are Hispanic; any individual who did not fall into this category was categorized as a non-Hispanic patient. Patient-specific characteristics are reported by hospitals to the PHD, but how these data were collected (i.e., patient reported) at each patient encounter is not available. For most of the data available, less than 1% of patient records have information that is missing. Other data elements, such as patient demographic and diagnostic information, have less than 0.01% missing data [39]. Facility characteristics included U.S. Census region, rural or urban location, teaching status, and bed size. Additional CDI-related characteristics included inpatient admission or outpatient encounter, admission type (emergency, urgent, elective), CDI diagnosis type (admitting, primary, secondary), severity indicators, and treatment patterns. Serum creatinine (SCr) and white blood cell (WBC) count values were obtained from the Premier general laboratory file and stratified according to the Infectious Diseases Society of America (IDSA)/Society for Healthcare Epidemiology of America (SHEA) guideline severity criteria if they occurred anytime during the encounter: SCr ≥ 1.5 mg/dL and WBC > 15 × 10^3^ cells/µL [22]. Patients with either severity criterion were classified as “severe CDI.” CDI therapies included administration of at least one dose of the following agents during the encounter: oral or intravenous metronidazole, oral vancomycin, fidaxomicin, bezlotoxumab, and FMT. These treatments were identified using Premier’s standard charge code for each therapy. Hospital-associated costs, or total encounter costs, were extracted from the patient cost variable provided by Premier, which represents the total cost to the hospital to treat the patient during the encounter (includes all supplies, labor, equipment, etc.). Patient-associated charges were the total charged amount of billed items during the hospital encounter. Recurrent CDI was defined as an additional encounter anytime following the initial CDI encounter that also included an ICD-10 code for CDI plus a positive *C. difficile* stool test. Lastly, for hospitalized patients only, in-hospital mortality was identified based on a discharge disposition of “expired.” This represents all-cause mortality. Hospital LOS was captured as a continuous variable and also dichotomized as ≤7 days and >7 days as 7 days was the median LOS for all patients included in the study.

### 4.5. Statistical Methods

Data and statistical analyses were conducted using JMP Pro 16^®^ (SAS Institute, Cary, NC, USA). Patient characteristics of the index and recurrence encounters were first presented descriptively and compared using the chi-square or Wilcoxon rank sum test. Next, CDI treatments, costs, charges, and other health outcomes were compared between each patient group by race (Black, other, White), ethnicity (Hispanic, non-Hispanic), sex (male, female), and age group (less than 18 years, 18 to 64 years, and 65 years and older) using the chi-square, Fisher’s exact, Wilcoxon rank sum, or Kruskal–Wallis test as appropriate. The race category “other” was used for comparison as the numbers from some race categories were too small for meaningful analyses. Next, each group was assessed as an independent predictor of health outcomes in a series of multivariable logistic regression models, including the following covariates: age group, sex, race, ethnicity, payor, region, urban hospital status, teaching hospital status, hospital bed size, CDI type (admitting/primary/secondary), CDI treatments (metronidazole, vancomycin, fidaxomicin), and CDI severity (except for the severity outcome). All tests were two-sided and performed at the 5% level of significance.

## 5. Conclusions

In this nationally representative study utilizing the Premier Healthcare Database, disparities were present in CDI treatment and related outcomes between groups defined age group, sex, race, and ethnicity. The use of certain guideline-recommended therapies was low in all patient subgroups. Overall, age 65+ years and 18 to 64 years, male sex, Black race, other race, and Hispanic ethnicity were predictors of severe CDI in index and recurrent cases. Male sex was also a predictor of all-cause, in-hospital mortality and Black race and male sex were associated with prolonged hospital stays for index encounters. These subgroups were not independently associated with all-cause, in-hospital mortality or hospital LOS in recurrent cases. While several determinants of health may play a role in our observations, this study suggests the need for increased efforts in providing equitable patient care in order to improve CDI-related outcomes and decrease overall healthcare burden and expenditures.

## Figures and Tables

**Table 1 antibiotics-11-01203-t001:** Baseline characteristics of CDI encounters.

Characteristic	Index Encounters (*n* = 38,764)	rCDI Encounters (*n* = 6567)	*p*-Value
Age, median (IQR)	68 (56–78)	69 (56–79)	0.009
Female sex, %	57.6	62.1	<0.001
Race, %			0.003
Black	12.3	12.8	
Other	6.6	5.3	
White	81.1	81.9	
Hispanic ethnicity, %	5.6	4.8	0.023
Payor, %			<0.001
Medicare	64.5	69.5	
Medicaid	11.4	12.4	
Managed care	14.2	10.8	
Commercial	3.9	3.2	
Indigent/charity/self-pay	3.4	2.3	
Other	2.6	1.8	
US Census region, %			<0.001
Midwest	23.1	25.6	
Northeast	13.6	12.3	
South	60.2	59.2	
West	3.1	2.9	
Inpatient admission, %	87.2	80.9	<0.001
Admission type, %			<0.001
Emergency	78.3	77.4	
Urgent	10.3	7.9	
Elective	9.3	12.4	
Trauma	0.5	0.1	
Unknown	1.6	2.2	
CDI diagnosis type, %			<0.001
Admitting	10.0	19.5	
Primary	23.4	26.1	
Secondary	66.6	54.4	
CDI present on admission, %	59.7	63.9	<0.001
Teaching hospital, %	46.1	45.5	0.391
Urban hospital, %	83.5	81.0	<0.001
Hospital bed size, %			<0.001
000–099	7.5	7.6	
100–199	14.0	14.6	
200–299	16.5	16.7	
300–399	18.5	17.1	
400–499	12.1	14.6	
500+	31.4	29.4	

rCDI = recurrent *Clostridioides difficile* infection.

**Table 2 antibiotics-11-01203-t002:** CDI treatment patterns by patient subgroups for index encounters.

	**Age Group**				**Sex**		
	**<18 Years** **(*n* = 486)**	**18–64 Years** **(*n* = 15,831)**	**65+ Years** **(*n* = 22,447)**	***p*-Value**	**Female** **(*n* = 22,330)**	**Male** **(*n* = 16,433)**	***p*-Value**
CDI therapies, %							
Metronidazole	46.5	43.6	44.3	0.230	44.4	43.7	0.208
Vancomycin	37.9	74.8	76.2	<0.001	74.0	76.7	<0.001
Fidaxomicin	0.1	4.4	4.6	<0.001	4.6	4.3	0.087
Bezlotoxumab	0.0	0.1	<0.1	0.721	<0.1	<0.1	0.698
FMT	0.0	0.1	0.1	0.502	0.1	0.1	
Costs, median (IQR)							
Hospital costs	5407	10,508	11,811	<0.001	9999	13,586	<0.001
	(1752–20,360)	(4768–28,124)	(5892–25,572)		(4901–22,281)	(6152–32,581)	
Patient charges	16,249	42,612	45,333	<0.001	39,387	52,321	<0.001
	(6710–57,137)	(20,188–10,684)	(23,347–95,867)		(19,963–83,200)	(24,622–12,247)	
	**Race**				**Ethnicity**		
	**Black** **(*n* = 4705)**	**Other** **(*n* = 2507)**	**White** **(*n* = 30,898)**	***p*-Value**	**Hispanic** **(*n* = 1837)**	**Non-Hispanic** **(*n* = 30,801)**	***p*-Value**
CDI therapies, %							
Metronidazole	42.5	48.9	44.0	<0.001	48.9	44.6	0.003
Vancomycin	74.0	77.6	75.3	0.003	78.0	74.9	0.003
Fidaxomicin	4.1	3.9	4.5	0.214	5.3	4.7	0.287
Bezlotoxumab	<0.1	<0.1	<0.1	0.992	0.0	0.1	0.621
FMT	0.1	0.1	0.1	0.761	0.4	0.1	0.010
Costs, median (IQR)							
Hospital costs	13,985	15,162	10,672	<0.001	11,154	10,959	0.013
	(6650–34,557)	(6649–37,911)	(5103–24,406)		(5415–27,878)	(5248–25,384)	
Patient charges	53,811	56,681	41,817	<0.001	47,957	42,888	<0.001
	(26,609–128,430)	(25,457–144,071)	(20,894–91,677)		(25,082–112,535)	(21,503–94,465)	

CDI = *Clostridioides* difficile infection; FMT = fecal microbiota transplantation; IQR = interquartile range.

**Table 3 antibiotics-11-01203-t003:** CDI treatment patterns by patient subgroups for recurrent CDI encounters.

	**Age Group**				**Sex**		
	**<18 Years** **(*n* = 62)**	**18–64 Years** **(*n* = 2613)**	**65+ Years** **(*n* = 3892)**	***p*-Value**	**Female** **(*n* = 4078)**	**Male** **(*n* = 2489)**	***p*-Value**
CDI therapies, %							
Metronidazole	32.3	35.9	36.3	0.770	35.9	36.3	0.787
Vancomycin	64.5	68.4	69.2	0.596	66.2	73.2	<0.001
Fidaxomicin	1.6	13.7	14.8	0.001	14.5	13.8	0.466
Bezlotoxumab	0.0	0.0	0.3	0.009	0.2	0.2	0.774
FMT	0.0	1.3	1.5	0.829	1.3	1.4	0.745
Costs, median (IQR)							
Hospital costs	9674	8909	9126	0.359	8401	10,284	<0.001
	(2590–36,833)	(4399–18,147)	(4319–17,748)		(3889–16,476)	(4989–20,446)	
Patient charges	26,990	34,637	32,311	0.002	30,919	37,563	<0.001
	(7374–95,676)	(17,262–68,846)	(15,655–63,973)		(14,813–61,165)	(18,388–74,918)	
	**Race**				**Ethnicity**		
	**Black** **(*n* = 831)**	**Other** **(*n* = 344)**	**White** **(*n* = 5312)**	***p*-Value**	**Hispanic** **(*n* = 460)**	**Non-Hispanic** **(*n* = 5589)**	***p*-Value**
CDI therapies, %							
Metronidazole	35.4	44.8	35.6	0.003	43.7	36.4	0.002
Vancomycin	73.8	75.6	67.7	<0.001	80.4	68.1	<0.001
Fidaxomicin	13.1	10.8	14.6	0.076	10.4	15.0	0.006
Bezlotoxumab	0.1	0.0	0.2	1.000	0.0	0.2	0.617
FMT	0.8	0.6	1.5	0.161	1.7	1.4	0.533
Costs, median (IQR)							
Hospital costs	12,352	11,532	8502	<0.001	11,019	9005	<0.001
	(6193–25,140)	(5490–25,210)	(3918–16,636)		(5658–25,542)	(4340–17,876)	
Patient charges	47,626	43,389	31,316	<0.001	48,337	33,488	<0.001
	(24,673–97,979)	(21,452–88,595)	(14,645–60,680)		(26,220–103,950)	(16,482–65,912)	

CDI = *Clostridioides* difficile infection; FMT = fecal microbiota transplantation; IQR = interquartile range.

**Table 4 antibiotics-11-01203-t004:** CDI outcomes by patient subgroups for index encounters.

	Severe CDI		CDI Recurrence ^c^		In-Hospital Mortality ^a^		Hospital LOS > 7 Days ^a^	
	%	aOR (95% CI) ^b^	%	aOR (95% CI) ^b^	%	aOR (95% CI) ^b^	Median (IQR)	aOR (95% CI) ^b^
Age group								
65+ years	45.9	1.99 (1.40–2.81)	10.4	2.07 (1.11–3.88)	7.5	2.55 (0.79–8.24)	7 (4–13)	1.01 (0.70–1.46)
18–64 years	36.6	1.80 (1.28–2.54)	9.1	2.01 (1.09–3.73)	4.5	1.64 (0.79–5.28)	7 (4–14)	0.87 (0.61–1.25)
<18 years	18.3	1.00 (reference)	8.1	1.00 (reference)	1.3	1.00 (reference)	5 (3–14)	1.00 (reference)
Sex								
Male	47.0	1.45 (1.37–1.53)	9.3	1.00 (0.91–1.10)	7.2	1.17 (1.05–1.31)	8 (4–15)	1.25 (1.18–1.32)
Female	37.8	1.00 (reference)	10.2	1.00 (reference)	5.4	1.00 (reference)	6 (4–12)	1.00 (reference)
Race								
Black	53.5	1.85 (1.71–2.00)	10.5 ^d^	1.17 (1.03–1.34)	7.1	0.96 (0.82–1.13)	8 (5–16)	1.29 (1.19–1.41)
Other	45.7	1.26 (1.12–1.41)	8.7 ^d^	0.91 (0.73–1.12)	7.8	0.97 (0.77–1.24)	8 (4–16)	1.06 (0.93–1.21)
White	39.6	1.00 (reference)	9.8	1.00 (reference)	5.9	1.00 (reference)	7 (4–13)	1.00 (reference)
Ethnicity								
Hispanic	44.2	1.22 (1.08–1.38)	9.0 ^d^	0.94 (0.76–1.17)	5.6 ^d^	0.93 (0.72–1.21)	7 (4–14) ^d^	0.89 (0.78–1.02)
Non-Hispanic	41.8	1.00 (reference)	10.0	1.00 (reference)	6.0	1.00 (reference)	7 (4–13)	1.00 (reference)

CDI = *Clostridioides* difficile infection; IQR = interquartile range. ^a^ Data represent hospitalized patients only. ^b^ Model covariates included: age group, sex, race, ethnicity, payor, region, urban hospital status, teaching hospital status, hospital bed size, CDI type (admitting/primary/secondary), CDI treatments (metronidazole, vancomycin, fidaxomicin), and CDI severity (except the severity outcome). ^c^ Data represent patients who survived the index visit in 2018–2020 only. ^d^
*p*-value for comparison between was >0.05 indicating non-significance compared reference group.

**Table 5 antibiotics-11-01203-t005:** CDI outcomes by patient subgroups for recurrent CDI encounters.

	Severe CDI		In-Hospital Mortality ^a,d^		Hospital LOS > 7 Days ^a^	
	%	aOR (95% CI) ^b^	%	aOR (95% CI) ^b^	Median (IQR)	aOR (95% CI) ^b^
Age group						
65+ years	45.9	11.98 (1.49–96.26)	4.7	1.05 (0.52–30.06)	6 (4–10) ^c^	0.86 (0.67–1.64)
18–64 years	36.8	9.65 (1.21–77.06)	3.4	1.00 (reference)	6 (4–11) ^c^	0.66 (0.30–2.65)
<18 years	8.0	1.00 (reference)	0.0	---	7 (4–22)	1.00 (reference)
Sex						
Male	48.7	1.62 (1.42–1.86)	4.6 ^c^	1.01 (0.72–1.41)	7 (4–12)	1.19 (1.03–1.38)
Female	37.5	1.00 (reference)	3.9	1.00 (reference)	6 (4–10)	1.00 (reference)
Race						
Black	56.4	2.12 (1.75–2.56)	5.2 ^c^	0.81 (0.52–1.1.30)	7 (4–14)	1.43 (1.16–1.75)
Other	51.7	1.55 (1.12–2.14)	5.1 ^c^	0.89 (0.43–1.84)	7 (4–12)	1.18 (0.84–1.64)
White	38.7	1.00 (reference)	3.9	1.00 (reference)	6 (4–10)	1.00 (reference)
Ethnicity						
Hispanic	55.2	1.98 (1.43–2.73)	6.3 ^c^	0.93 (0.46–1.89)	6 (4–12) ^c^	0.85 (0.60–1.19)
Non-Hispanic	41.7	1.00 (reference)	4.2	1.00 (reference)	6 (4–11)	1.00 (reference)

CDI = *Clostridioides* difficile infection; IQR = interquartile range. ^a^ Data represent hospitalized patients only. ^b^ Model covariates included: race, ethnicity, sex, age group, payor, region, urban hospital status, teaching hospital status, hospital bed size, CDI type (admitting/primary/secondary), CDI treatments (metronidazole, vancomycin, fidaxomicin), and CDI severity (except the severity outcome). ^c^
*p*-value for comparison between was >0.05 indicating non-significance compared reference group. ^d^ Patients <18 years old removed from mortality analysis.

## Data Availability

Restrictions apply to the availability of these data. Data were obtained from Premier, Inc., and are available from the authors with the permission of Premier, Inc.

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
