# Peer review of "Clostridioides difficile Infection Treatment and Outcome Disparities in a National Sample of United States Hospitals"

_antibiotics, 2022, doi:10.3390/antibiotics11091203_

Round 1

Reviewer 1 Report

This manuscript by Eric H. Young et al. describes clinical and demographic data in light of C. difficile infection in a U.S. national cohort.

The manuscript merits multiple edits and revisions before possible acceptance for publication.

Global: prefer passive rather than active turns of phrase. "et al." and "e.g." in italics.

Introduction: The first paragraph is far too broad/imprecise. Reword it.

Results: How did the authors consider duplicates? Even in the absence of recurrence (what is the precise definition anyway?), including the same patient twice is a major bias, and deserves recalculation of all data in light of this important change.

Table 1: Please compare the cohort to the full US population. P-values below the significance level are sufficiently precise, please do not report this number of decimal places.

Section 2.2: because these data could be biased due to changes in diagnostic methods and treatment recommendations, the authors should stratify their results by year of diagnosis.

Table 4/5: Because the authors identified parameters significantly associated with CDI (recurrent or not), they should consider producing a prognostic/diagnostic score.

Methods: Since some children carry C. difficile before the age of 3-5 years, not all isolation can be considered disease, please consider this information in your statistics.

Methods: How did the authors account for the bias associated with multiple testing? What multiple testing correction did they use?

Author Response

Thank you for your review. Please see our responses and associated manuscript changes attached. 

Reviewer 2 Report

Very interesting study, and well organized and explained. It shows one again the disparity in the health-care system.

Did you check the microbiota of those population ? As the alimentory regimen and his quality may very different between those population.

Author Response

(The authors gave the same response as above.)

Reviewer 3 Report

A great review that summarizes four stages of clinical trails of a very promising pharmaceutic.  

Author Response

(The authors gave the same response as above.)

Round 2

Reviewer 1 Report

The manuscript has been correctly revised according to my previous comments.